# Direct Patterning and Spontaneous Self-Assembly of Graphene Oxide via Electrohydrodynamic Jet Printing for Energy Storage and Sensing

**DOI:** 10.3390/mi11010013

**Published:** 2019-12-19

**Authors:** Bin Zhang, Jaehyun Lee, Mincheol Kim, Naeeung Lee, Hyungdong Lee, Doyoung Byun

**Affiliations:** 1Department of Mechanical Engineering, Sungkyunkwan University, 2066 Seobu-Ro, Jangan-Gu, Suwon 440-746, Korea; zhangbinyydi@gmail.com (B.Z.); Emmettbread@gmail.com (J.L.); nasakmc@gmail.com (M.K.); 2School of Advanced Materials Science & Engineering SKKU Advanced Institute of Nanotechnology (SAINT) and Sungkyunkwan University (SKKU), 2066 Seobu-Ro, Jangan-Gu, Suwon 440-746, Korea; nelee@skku.edu

**Keywords:** electrohydrodynamic jet printing, self-assembly, graphene oxide, ammonia sensor, supercapacitor

## Abstract

The macroscopic assembly of two-dimensional materials into a laminar structure has received considerable attention because it improves both the mechanical and chemical properties of the original materials. However, conventional manufacturing methods have certain limitations in that they require a high temperature process, use toxic solvents, and are considerably time consuming. Here, we present a new system for the self-assembly of layer-by-layer (LBL) graphene oxide (GO) via an electrohydrodynamic (EHD) jet printing technique. During printing, the orientation of GO flakes can be controlled by the velocity distribution of liquid jet and electric field-induced alignment spontaneously. Closely-packed GO patterns with an ordered laminar structure can be rapidly realized using an interfacial assembly process on the substrates. The surface roughness and electrical conductivity of the LBL structure were significantly improved compared with conventional dispensing methods. We further applied this technique to fabricate a reduced graphene oxide (r-GO)-based supercapacitor and a three-dimensional (3D) metallic grid hybrid ammonia sensor. We present the EHD-assisted assembly of laminar r-GO structures as a new platform for preparing high-performance energy storage devices and sensors.

## 1. Introduction

Recently, the self-assembly of two-dimensional (2D) materials has been considered a significant research area owing to its remarkable physical and chemical properties in various applications [1,2,3]. Among the 2D materials, layer-by-layer (LBL) laminar-structured graphene nanostructures exhibit superior electrical conductivity, mechanical strength, and chemical stability [4,5,6,7,8,9]. These promising properties make them an intriguing candidate for high-performance electronic devices such as graphene-based transistors, transparent electrodes, electrochemical sensors, and energy storage devices owing to the confined stacking and low contact resistance between interlayers [10,11,12]. Free-standing LBL graphene-based films can be obtained by solution processing, for example, spraying, dispensing, and dip-coating at room temperature [8,13]. These methods have advantages for the direct manipulation of the working parameters and LBL assembly of graphene nanostructures at both micro- and macroscopic scales [7]. However, the conventional methods are often limited because of low efficiency, limited electrical conductivity, and the geometry of the product. Also, to make sure graphene nanostructures align homogenously, a high temperature is often required, and the evaporation of the solvents takes a long time [10,13,14]. Also, for the dip-coating and conventional spraying, the geometry cannot be controlled efficiently and more complex processes may be required if one needs to apply the product into practical applications.

In terms of the direct patterning, the geometrical arrangement and the direct assembly of micro- and nano-scale material stimulated enormous interest in various technologies. They provide many potential applications in various fields such as chemical sensing, tissue engineering, energy harvest, and so on. Especially, the electric field provides us an efficient and reliable way. By utilizing the local surface charges and manipulating the electric field, several unconventional jet printing techniques including bipolar pyroelectrospinning, μ-pyro-electrospinning, and atomic force microscope charge writing have presented the capability of making direct patterns for the production of high-value structures [15,16,17]. P. Galliker et al. presented electrohydrodynamic (EHD)-jet and nanoscale 3D printing with feature sizes down to 50 nm [18]. They demonstrated the autofocussing phenomenon caused by local electrostatic field enhancement, resulting in large aspect ratio structure formation. These works inspired us to study the electric field for LBL graphene direct assembly. 

Many researchers have proposed the electrohydrodynamic (EHD) jet printing method for aligning one-dimensional (1D) nanomaterials along the jet streamline [19]. EHD jet printing utilizes proper electric fields, instead of piezoelectric or acoustic force, to induce the ejection of charged jets or droplets toward a substrate. By the compositive mechanism of the mechanical stretching of the charged jet and the shear effect caused by the velocity distribution, the EHD jet printing method can successfully align 1D nanowires over a large area. On the basis of this phenomenon, a high-resolution pattern (<1 μm) can be printed on various surfaces to fabricate micro/nanoscale printed electronics using functional or sacrificial inks efficiently [20,21]. Electric field-induced fluid dynamics also affect the geometry of the pattern of printed materials during the EHD-jet printing process. Previous reports also found that field-induced torque caused by the polarization of graphite flakes under a strong electric field could induce the orientation of the graphite flakes or graphene [22,23,24]. However, they only focused on the electromagnetic behavior of the 2D materials, and it would be difficult to expect practical application owing to the limited experimental method for patterning the 2D materials. To solve the bottleneck, an alternative should be found to realize LBL laminar-structured graphene nanostructures on various substrates by dominant convection fluid flow during printing, and sufficient electric charge to accelerate the solvent evaporation rate [25].

In this study, we investigated the theoretical and experimental mechanism of LBL self-assembly of graphene oxide (GO) during EHD jet printing. We found that hydrodynamically induced alignment occurs during the stretching of EHD jet fibers. We also defined the domination region of the electric field-induced orientation of the GO flakes. On the substrate area after jetting, the fluid flow during evaporation induced interfacial assembly of the GO flakes and formed ultra-uniform laminar nanostructures successfully. On the basis of the printed GO patterns, we reduced GO to reduced graphene oxide (r-GO) using the vaporized reducing agent, N_2_H_4_, hydrazine. Compared with the conventional direct dispensing method, the surface roughness and electrical conductivity of the EHD jet-printed LBL nanostructures were significantly improved. We further applied this technique to fabricate the r-GO-based ammonia sensor with a 3D metallic grid.

## 2. Materials and Methods 

### 2.1. Ink Preparation

The GO was prepared using a modified Hummers method from graphite flakes (Alfa Aesar, 325 mesh). Briefly, a mixture of graphite, NaNO_3_, H_2_SO_4_, and KMnO_4_ at a ratio of 1:1:60:4 was kept in an ice-water bath and stirred for 24 h. After that, H_2_SO_4_ (5 wt% aqueous solution) and H_2_O_2_ (30 wt% aqueous solution) were added at a ratio of 1:30 and further stirred for 1 h. To avoid nozzle clogging and to reduce the size of the GO flakes, the GO aqueous solution was treated with ultra-sonication for 5 min. The solution was then centrifuged at 2000 rpm for 5 min to decant the supernatant and discard the pellet. We only used the supernatant as the printing ink. The ink was kept in the refrigerator at 5 °C.

### 2.2. EHD Jet Printer

An EHD printer developed by Enjet Inc. (MX-model, Suwon, Korea) was used in this study. A controller inside the machine guides the movement of the nozzle along the *Z*-axis and the substrate in the X- and Y-axis. The maximum velocity and the acceleration of the horizontal stage (X- and Y-axis) are 2000 mm/s^2^ and 5000 mm/s^2^, respectively. The stainless nozzle (inner diameter: 100 μm) was used to dispense the continuous jet. For the counter substrate, an aluminum plate was connected to the ground. The contact angle of the printing substrate is 31.2°, which considers as the hydrophilic substrate. The nozzle was positioned 4 mm and 10 mm above the silicon wafer substrate and the counter electrode for direct dispensing and EHD jet printing, respectively. The high voltage was supplied by the function generator with an amplifier (TREK, 609E-6) ranging between 0 kV and 5 kV. To supply the ink, a syringe pump was used (Harvard, NEW PHD ULTRTM Nanomite, Holliston, United States). The ink flow rate was set to 8 μL/min.

### 2.3. Fabrication of Cell Type Supercapacitor

The printed r-GO was used as the electrode part for a coin cell type supercapacitor. First, the LBL GO was printed on two stainless steel spacer disks, which had a diameter and thickness of 16 mm and 1 mm, respectively. Then, we used a two-step chemical reduction method to obtain r-GO. The printed sample was first treated by hydrazine vapor at 50 °C for 18 h. Then, we cleaned the samples using deionized (DI) water. Finally, the sample was heated at 250 °C in a vacuum chamber to remove the remaining hydrazine and DI water. For the solid-state electrolyte, we followed the synthesis method introduced by Sung et al. [26]. The polyvinyl alcohol (PVA) powder (average Mw ~13,000, Sigma Aldrich) was dissolved in DI water with the mixing ratio of 5 wt%, and was then stirred vigorously on a hotplate (> 95 °C) until it melted and became a transparent solution. After dissolving all the PVA powder in the DI water, the solution was cooled to room temperature, and 0.1 M H_3_PO_4_ (85%) aqueous solution was added and stirred for 5–6 h for homogeneity of the H_3_PO_4_ in PVA gel solution at room temperature. As the last step, the PVA/H_3_PO_4_ gel solution was poured on a prepared commercial polypropylene (PP) sheet, and was then kept on a 30–40 °C hot-plate for approximately 1 d to evaporate the water compounds. After the evaporation process was finished, the PVA/H_3_PO_4_ polymer electrolyte sheet was punched to prepare a circle-shaped layer (18 mm diameter). With the two prepared r-GO electrodes and PVA/H_3_PO_4_ polymer electrolyte, the coin cell was assembled with the conventional assembly procedure of the 2032-coin cell. The laminated r-GO membranes were stacked face-to-face within the coin cell with the separator making direct contact. The assembled coin cell was stored for 2–3 h to stabilize the material homogeneity before electrochemical measurement.

### 2.4. Preparation of 3D Metallic Grid for Gas Sensor

Before fabricating the gas sensor application, the metallic grid was prepared by using the EHD jet printer and commercialized nano-ink, which is composed of silver nanoparticles (Ag NPs) and polymer binder (EM-SP01, Enjet Inc.). To generate a continuous ejection of high-viscous ink containing Ag NPs, a direct current (DC) voltage of 2.5 kV was applied to a nozzle (inner diameter: 100 μm). The aspect ratio of printed geometry was controlled by the number of printing layers. For a low aspect ratio sample, we tried to overprint ten times, and sixty times for the high aspect ratio sample. The measured aspect ratios were approximately 5 and 25 for the low and high aspect ratio samples, respectively. After the printing process, the 3D metallic grid was sintered in an oven at 200 °C for 30 min. We then overprinted again twenty times with GO ink on the metallic grid. Finally, we carried out the two-step chemical reduction method to obtain the r-GO pattern.

### 2.5. Characterizations

The surface roughness and morphology of the printed geometry were observed using a 3D profiler (NanoView Inc., Daejeon, Korea) and field-emission scanning electron microscopy (FESEM) (JEOL, JSM-7600F, Tokyo, Japan). The electrical resistance of the printed GO and r-GO was measured using a probe-station (MS tech. Model 5500, Hwaso, Korea) and a source meter (Keithley 2400 series, Cleveland, United States). The supercapacitor was tested using the charge/discharge system (WonATech, Seoul, Korea). Cyclic voltammetry (CV) and galvanostatic charge/discharge (GCD) were measured using a potentiostat (Metrohm Autolab, PGSTAT302N model, Utrecht, Netherlands). A sealed symmetrical coin cell (CR2032) was used for all electrochemical measurements. CV was performed at various scan rates ranging from 20 to 100 mV/s. The performance of an ammonia sensor was measured in a gas sensing system with a gas supply and control units [27]. The temperature inside the chamber was stabilized at 27 °C. Dry synthetic air (21% *v/v* O_2_/N_2_) was used for purging and dilution. The maximum flow rate (500 sccm) of the carrier gas was maintained. The maximum concentration of the 500 sccm testing gas was 100 ppm for NH_3_ in air and was controlled by mass flow controllers (MFCs). During gas flow, electrical currents were monitored using a source meter (Keithley 2400 series, Cleveland, United States). The implemented software (Keithley) allowed for the control of the bias voltage (2 V), limiting the current (10 mA), and recording the output signal of the current and resistance.

## 3. Results

### 3.1. EHD Jet Printing and Self-Assembly of GO

As shown in Figure 1a, we carried out the EHD jet printing process using DI water-based GO ink. When we applied a high voltage between the nozzle and the substrate, the Taylor-cone jet formed at the nozzle tip owing to the tangential electric shear stress acting on the surface of the liquid (Figure 1b) [19]. Using successive jetting, we induced the self-assembly of GO both inside the liquid jet and on the substrate.

To understand this phenomenon in detail, we considered three mechanisms that are related to the flow-induced alignment and assembly process, as shown in Figure 1c–e. In the initial state, the GO flakes are randomly dispersed in water because of their oxygen functionalities. After jetting from the nozzle tip, the diameter of the liquid jet decreases to several microns owing to electrical stretching against the surface tension of the liquid. Also, owing to the tangential electric stress, the maximum velocity can be obtained at the air/liquid interfaces. We assume that the shear effect caused by the velocity distribution of the liquid jet near the nozzle tip is one of the critical factors. Inside the liquid jet, dispersed nanomaterials are affected by drag force. The hydrodynamical drag force can be defined as follows [19,28]:(1)$$T_{V}=\frac{4\pi \eta \mu a^{2}}{b\left(\ln\left(\frac{4a}{b}\right)-0.72\right)}\sin\theta \;,$$
where *η* and *μ* are the fluid viscosity and velocity, respectively. For the DI water, *η* ~ 1.006 mPa∙s at 20° C. a and b represent the length and thickness of a single GO flake, respectively. Herein, we assume that *a* = *b* = 1 μm. *θ* is the angle between the electric field direction and the semi-major axis of the GO flake. Just before deposition on the substrate, the jet fluid travels forward to the substrate, as shown in Figure 1d. The GO flakes can be further oriented along with the electric flux in the strong electric field in effect between the nozzle tip and the substrate. We assume that *θ* is the angle between the electric field direction and the semi-major axis of the GO flake. Thus, the hydrodynamic drag force is a function of *θ*. Meanwhile, the electric field-induced torque acting on single GO flake was estimated with the following equations [22,29]:(2)$$T_{E}=\frac{4\pi a^{3}}{3}\frac{\varepsilon }{\left(\frac{\pi }{2}-\frac{b}{a}\right)}E^{2}\sin2\theta \;,$$
(3)$$T_{E}=\frac{V}{2}\varepsilon_{0}\varepsilon_{2}\left(\frac{{\left(\sigma_{1}-\sigma_{2}\right)}^{2}}{\sigma_{1}\sigma_{2}}\right)E^{2}\sin2\theta \;,$$
(4)$$T_{E}=T_{V}\frac{4a^{2}b\varepsilon }{3\eta \mu }\frac{\left(\ln\left(\frac{4a}{b}\right)-0.72\right)}{\left(a\pi -2b\right)}E^{2}\cos\theta ,$$
where *E* is the local electric field between the nozzle and the substrate; *ε* is the permittivity; *ε*_0_ is the permittivity of free space; *ε*_2_ is the relative dielectric constant of the solvent (DI water, *ε*_2_ = 80.2); and *σ*_1_ and *σ*_2_ are the conductivity of the GO flake and DI water, respectively. *σ*_1_ ≈ 10^4^ S/m and *σ*_2_ ≈ 5.5 × 10^−6^ S/m. *V* is the volume of a single GO flake and *V* = *a*^2^*b*. If we assume *θ* = 45°, *T_V_* ≈ 4.868 × 10^−6^ N∙m and *T_E_* ≈ 3.005 × 10^−4^ N∙m. *T_E_* >> *T_V_*. Therefore, during this period, the electric field has a dominant effect on the GO alignment. After the ink was deposited on the substrate, the GO flakes pre-aligned along the printing direction. The ink possessed a sufficiently high storage elastic modulus to maintain the printed filamentary shape owing to the surface tension. Upon solvent evaporation, the viscosity of the ink was higher and further resulted in a pinning state at the gas−liquid−solid three-phase contact line [13,30]. The ink underwent longitudinal contraction. GO sheets were driven by the upward force of the evaporating flow to assemble at the air/water interface (Figure 1e). Meanwhile, the interfacial tension of the water compound compressed the sheets in the vertical direction as rapid evaporation occurred owing to the strong electrical charge. At this stage, the compressive force played a key role in forcing the GO sheets into alignment. The sheets gradually evolved into a more consistent arrangement because of the *π–π* conjugation effect and hydrogen bonding interaction [31]. The closely packed and laminar-structured GO lines were finally realized by the flow-induced alignment/orientation and evaporation-induced interfacial assembly hybrid process. Compared with other pool alignments of the GO product, which were fabricated using other methods (Figure 1f), our EHD-jet printing technique can significantly improve the GO flakes (Figure 1g).

### 3.2. Optimization of Printing Parameters

To achieve design pattern resolution and LBL GO nanostructures, it is important to optimize various printing conditions, such as the applied voltage, tip-to-the-collector distance, flow rate, and printing speed. Figure 2 presents the effect of primary parameters on the pattern width (i.e., resolution) and thickness. As expected, the resolution improves when the printing speed is increased (Figure 2a). In general, the optimal printing speed significantly improved the resolution and showed a linear relationship with the pattern resolution. The faster printing speed resulted in a smaller amount of ink deposited on the substrate. When the printing speed reached a critical value (> 200 mm/s), the pattern was disconnected instead of continuous. The optimal printing speed was found to be approximately 100 mm/s. Figure 2b also shows the effect of the applied voltage on the pattern width. The jetting was initiated at 1.8 kV and was stably maintained as the applied voltage increased to 4 kV. Smaller droplets and higher resolution of patterning were achieved by increasing the applied voltage. However, the width of the graphene pattern showed saturation at voltages higher than 3 kV. When the voltage was over 4 kV, the liquid could be sprayed at the apex of the meniscus and atomized into small droplets. Therefore, one should maintain the voltage between 2 and 4 kV to achieve stable jetting for the design pattern width. A higher voltage extracts more liquid from the nozzle meniscus with a strong electrical force, so the liquid supply rate is quite important for stable and reliable patterning. The voltage is also the key factor that affects the thickness of the jet fiber. As shown in Appendix A, the voltage increased from 2 kV to 3.5 kV. The jetted fiber thickness was decreased from 25 μm to 15 μm, approximately. Figure 2c shows the effect of flow rate on the pattern width. The pattern resolution is quite nonuniform when the flow rate is less than 10 μL/min. The optimal range of flow rate is between 10 and 14 μL/min. The pattern width could also gradually increase owing to the larger amount of dispensing liquid from the nozzle. The working distance between the nozzle and the substrate was fixed at 10 mm to provide enough space for jet stream formation and solvent evaporation. LBL GO nanostructures with different thicknesses were also deposited with multiple printing where the number of printed layers ranged from 10 to 60, as shown in Figure 2d. The inset photograph captured by SEM shows that the thickness linearly increases with the number of printed layers from 10 to 60. The multiple layers of printing allow for control of the pattern thickness as well as self-assembly of the GO flakes. Furthermore, the easy control of pattern thickness can be applied in various applications that require different thicknesses of graphene. The color of the pattern also shows a gradual change when the thickness is controlled. More information of the coffee ring effect and color change is shown in Appendix A.

### 3.3. Morphology and Electrical Conductivity of GO Laminar Structure

Figure 3 shows a 3D profile and SEM images to further demonstrate the surface morphology improvement from the EHD-jet printing technique. As mentioned in the introduction section, spraying, dispensing, and dip-coating was utilized to fabricate the LBL graphene structure. However, only the direct dispensing and EHD-jet printing belong to the nozzle-based printing techniques, which means the final pattern can be easily controlled compared with other methods. Thus, we only chose the direct dispensing as the reference method. The applied voltage was set to 0 kV. We maintained the other printing parameters. As a result of dispensing the GO suspension, we observed inhomogeneous morphologies because the deposited structure was governed by only the solvent evaporation process. Also, it was limited to reducing the pattern width during dispensing. Therefore, the coffee-ring effect and a low evaporation rate occurred (Figure 3a–c). On the other hand, EHD jet printing provided both flow-induced alignment and a fast evaporation process with high resolution. As shown in Figure 3d–f, we obtained a very uniform pattern over a large area, and the surface roughness was much lower than those printed by direct dispensing. From the SEM images, it is easy to compare the microscopic morphologies of the printed GO patterns. For the direct dispensing printed samples, the GO wrinkles on the surface were observed to result in high surface roughness (Figure 3g). The wrinkles remained even when multilayer printing was conducted, as shown in Figure 3h. Many defects can be seen when we look at the cross section (Figure 3i). GO flakes are not well aligned, and the wrinkle edge indicates a very high contact resistance between the graphene layers. Therefore, we concluded that direct dispensing is not suitable for laminar structure formation. From an overall perspective, the EHD printed GO pattern shows continuous lines with smooth edges and a flat surface morphology. The printed patterns show regular distribution on the surface area, and GO wrinkles, which are caused by constriction and overlapping of the GO sheets during the drying process, do not occur (Figure 3j,k). The laminar structure in the cross section further proves the alignment and dense stacking of the GO flakes (Figure 3l). We concluded that the fluid flow and electric field-induced alignment contributed to the GO laminar structure formation and significantly improved both the surface and inner morphology. 

To investigate the electrical properties of the LBL nanostructures, we prepared r-GO patterns from laminar-structured GO. Typically, the contact resistance between r-GO layers affects the electrical conductivity. Uniform and densely stacked r-GO would have a higher conductivity owing to the large contact area compared with irregularly aligned samples. As shown in Figure 4, a comparison of the electrical resistivity of the EHD-jet and direct dispensing printed samples was conducted. The conductivity of the GO patterns was 7.62 × 10^−2^ S/m. Obviously, for the printed r-GO pattern, the electrical conductivity strongly depends on the printing methods. In the case of dispensing, we calculated an average conductivity of approximately 7.03 × 10^3^ S/m, while the EHD-jet-printed r-GO pattern exhibited an electrical conductivity of 2.01 × 10^4^ S/m. This is an impressive result because the conductivity is among the highest reported in printed graphene electronics [32,33,34]. We expected that the conductivity value would be further enhanced by optimizing the size and distribution of GO for the ink. The EHD-jet printing significantly improved the conductivity.

### 3.4. Supercapacitor

To test the electrochemical performance of our printed layer-by-layer graphene as an electrode for supercapacitors, a symmetric two-electrode coin cell was assembled (Appendix A). Galvanostatic charge/discharge testing results are also shown in Appendix A. The constant current ranged from 25 μA to 100 μA. The scan rate is kept constant at 5 mV/s. It shows good response when we charge and discharge the printed graphene with different current density. Cyclic voltammetry (CV) is also performed as shown in Appendix A. The CV profiles of the various scanning rates range from 20 mV/s to 100 mV/s. All curves exhibit a nearly rectangular shape, with no largely apparent redox peaks associated with pseudo capacitance. A high specific capacitance of 96 F/g at a current density of 0.25 A/g in our electrodes confirms the very good conducting network of the graphene film with a large specific surface area. However, as we are using an aqueous solution electrolyte, arising from residual functional groups and defects could also be contributing to the observed capacitance. So, the performance needs to improve and the CVs need to be made more rectangular in future works. It is believed that electrodes with a layer-by-layer graphene play an important role in the capacitive performance of the device.

### 3.5. Fabrication of 3D Ammonia Sensor by EHD Jet Printing

A 3D ammonia sensor is presented by combining high aspect ratio 3D metallic grids and an LBL r-GO nanostructure. We demonstrated that the sensing area could be dramatically increased by high aspect ratio metallic grids (i.e., a 3D substrate). A schematic of the sensor fabrication process is shown in Figure 5a. First, the metallic grid was prepared using EHD-jet printing with Ag NP ink. We then performed GO printing on top of the 3D substrate. Finally, chemical reduction was conducted to obtain the r-GO for sensing. Figure 5b,c show the top and cross-section view of the sensor structure. Interestingly, the 3D metal grid was entirely covered with a free-standing LBL r-GO film. A magnified SEM image of the ultra-uniform aligned graphene laminar structure is shown in Figure 5d. The response behaviors and sensitivity of the sensors are indicated as the relative change of electrical resistance. The sensor response was then determined by measuring the resistance change in the presence of ammonia gas. Graphene usually behaves like a p-type semiconductor. Hence, its exposure to ammonia (an n-type dopant for graphene) led to the decrease in the resistance, accompanied by a decrease in the concentration of holes [35]. We assume the relative resistance change ΔR (%) = (R−R_0_)/R_0_ ×100%, where R_0_ is the initial resistance of the sensor in dry air before ammonia exposure and R is the resistance measured during the exposure. Figure 6a,b show the 3D sensor with different aspect ratios. Figure 6a shows the resistance variation during ammonia detection for a sensor with an aspect ratio of 5. The relative resistance change was approximately 15% when the sample was exposed to 100 ppm NH_3_ for 5 min. For a sensor with an aspect ratio of 25, the resistance change could reach over 70% owing to the large reaction area (Figure 6b). To further test the performance of the high aspect ratio ammonia sensor, which is shown in Figure 6b, we conducted the response upon exposure to different concentrations of ammonia ranging from 10 ppm to 50 ppm (Figure 6c). To demonstrate the rapid response to ammonia, the sensing performance during two successive cycles of exposure to ammonia for 2 min and airflow for 3 min is shown in Figure 6d. The saturation curve showing the on/off ratio is shown in Appendix A. The saturation happened after 10 min under exposure (100 ppm). The ΔR (%) is over 80%. This indicates we successfully applied the r-GO printing to the 3D substrate, which is a promising technique for fabricating high-performance chemical sensors compared with the previous works [35,36,37].

## 4. Conclusions

In summary, we presented the self-assembly of LBL GO via the EHD-jet printing technique. We found that both flow- and electric field-induced orientation that occurs along the jetting stream dominated the GO flakes alignment. Also, the evaporation-induced interfacial assembly that occurs at the water/air interface successfully assembled the GO flakes and formed ultra-uniform laminar structures on the substrate. Closely-packed GO lines with a perfectly ordered film structure were achieved. The surface roughness and electrical conductivity of the LBL structure were also significantly improved. We further applied this method in fabricating an r-GO-based supercapacitor and a 3D metallic grid hybrid ammonia sensor. The presented method for r-GO laminar structure fabrication provides a promising path towards high-performance energy storage devices and sensors.

## Figures and Tables

**Figure 1 micromachines-11-00013-f001:**
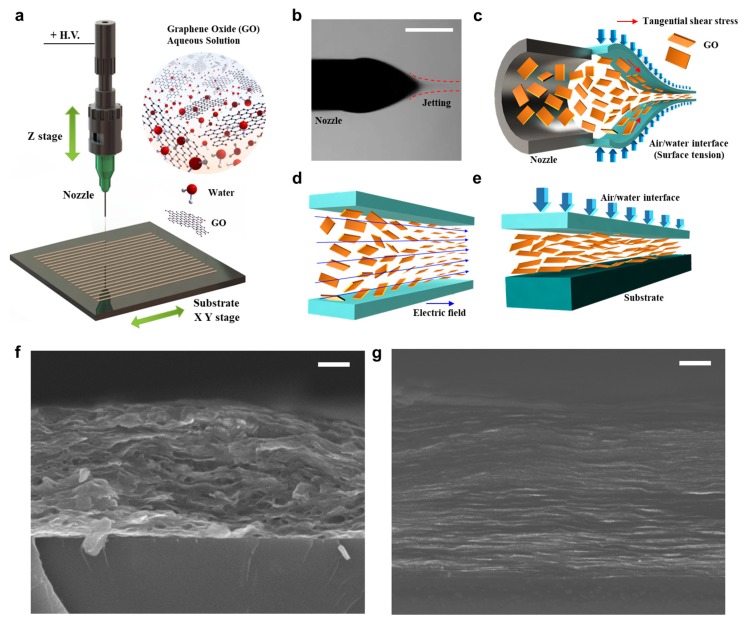
Schematic illustration of the electrohydrodynamic (EHD)-jet printing and alignment of the graphene oxide (GO). (**a**) The EHD printing system (X-model, www.enjet.co.kr), in which GO aqueous solution serves as the ink. (**b**) The CCD camera captures the stable cone jet. The scale bar is 200 μm. (**c**) A Taylor-cone jet forms at the nozzle tip owing to the tangential electric shear stress acting on the surface of the air/water interface. The jet fluid diameter decreases to several microns. Hydrodynamically induced alignment occurs during the acceleration/stretching process to homogenize the orientation order along the flow direction, against the random distribution of the initial state. (**d**) GO flakes oriented along the electric field in effect between the nozzle tip and substrate. (**e**) GO flakes are assembled at the air/water interface by the upward force of the evaporating flow. (**f**) Pool alignment of GO product using the dispensing method. (**g**) GO flakes prepared by the EHD-jet printing technique.

**Figure 2 micromachines-11-00013-f002:**
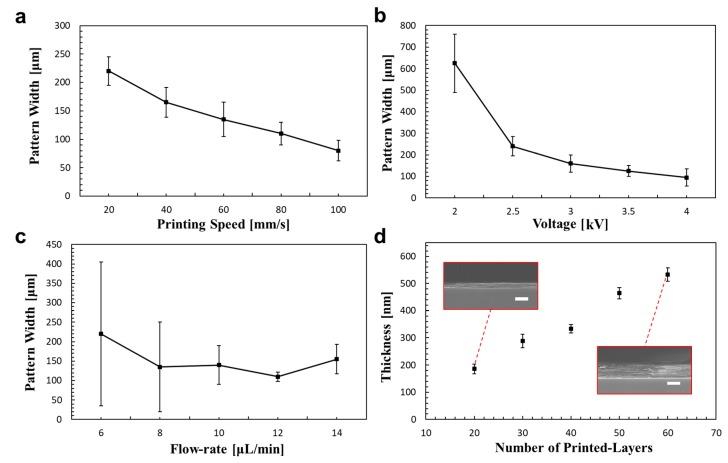
Effect of parameters on the pattern width. (**a**–**c**) Effect of printing speed, voltage, and flow rate on pattern width, respectively. (**d**) Relationship between pattern thickness and the number of printed layers. The scale bars in the insert are 500 nm.

**Figure 3 micromachines-11-00013-f003:**
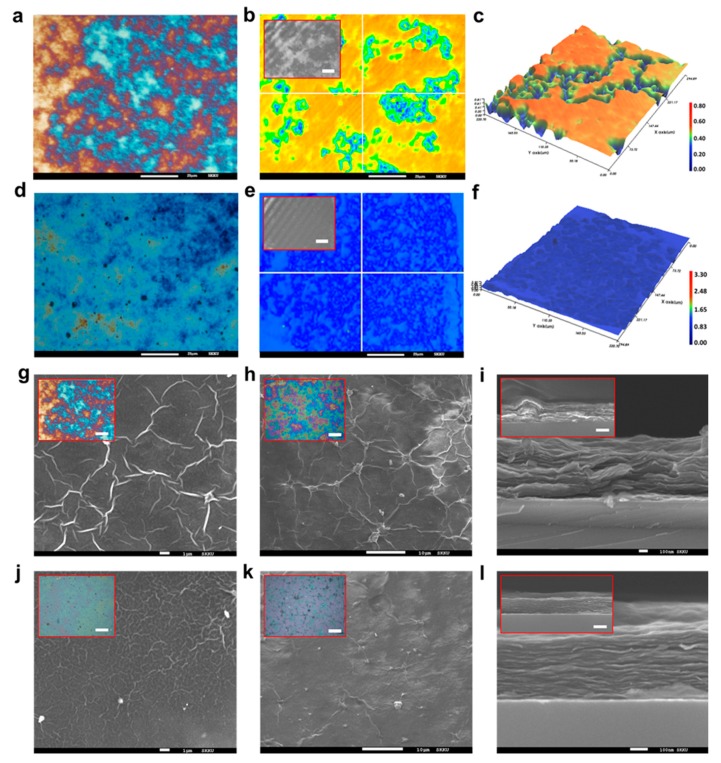
Microscopic morphologies of the printed GO patterns. (**a**–**c**) Microscope photo and 3D profile of direct dispensing fabricated sample. All the scale bars are 25 μm. (**d**–**f**) EHD jet-printed samples. All the scale bars are 25 μm. (**g**–**i**) Scanning electron microscopy (SEM) images show microscopic morphologies of the irregular aligned graphene surface and cross section. The scale bar is 1, 10, and 10 μm, respectively. (**j**–**l**) EHD jet printing samples. The scale bar is 1, 10, and 10 μm, respectively. All the scale bars in the inset are 25 μm and 500 nm for the microscope and SEM image, respectively.

**Figure 4 micromachines-11-00013-f004:**
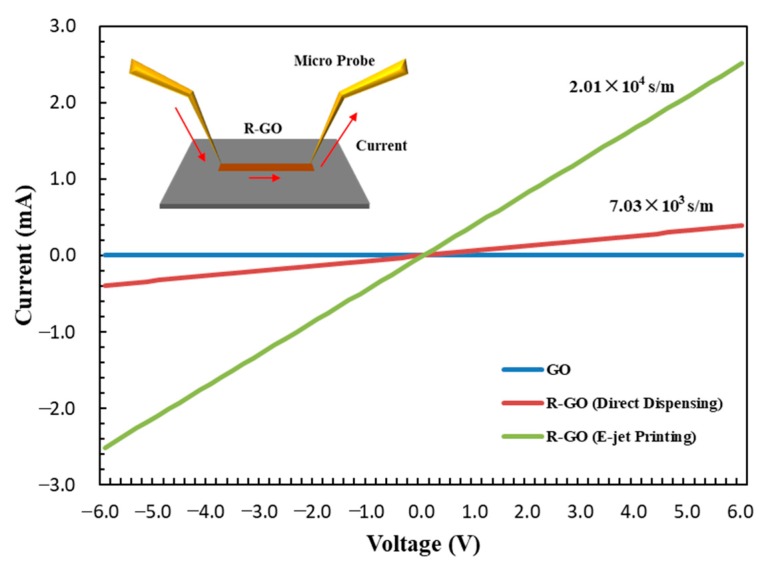
Comparison of the r-GO electrical conductivity printed by EHD-jet printing and direct dispensing (without electric field).

**Figure 5 micromachines-11-00013-f005:**
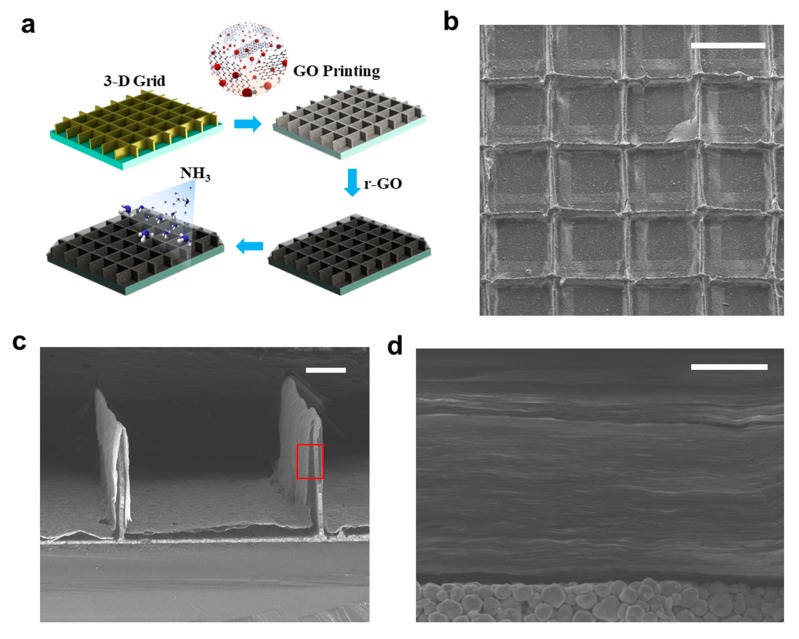
Structure of a 3D graphene-based ammonia sensor. (**a**) Fabrication process of a graphene ammonia sensor based on a high aspect ratio 3D metallic grid. (**b**) SEM image shows the top view of the sensor. The scale bar is 500 μm. (**c**,**d**) SEM image shows cross section of 3D metal grid covered with free-standing graphene film and magnified graphene laminar structure (marked as red region). The scale bar is 100 μm and 1 μm, respectively.

**Figure 6 micromachines-11-00013-f006:**
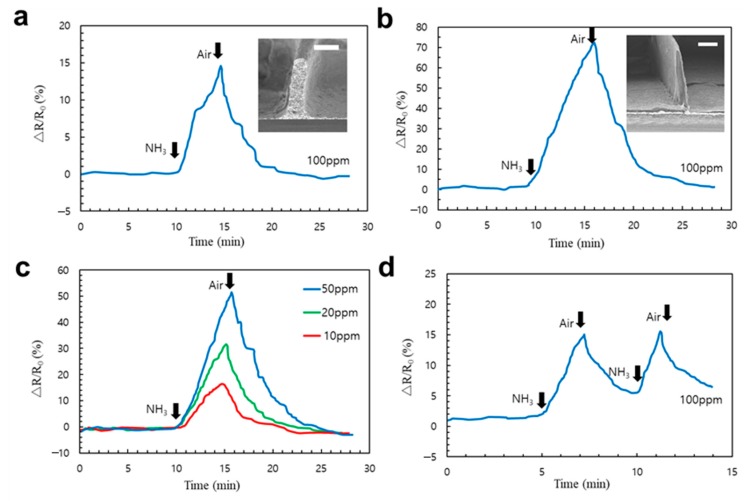
Resistance variation during ammonia detection. (**a**,**b**) 3D sensor with different aspect-ratio response upon exposure to ammonia (100 ppm). The aspect ratio is 5 and 25, respectively. The scale bar in the inset is 10 and 100 μm, respectively. (**c**) 3D ammonia sensor response upon exposure to different concentrations of ammonia ranging from 10 ppm to 50 ppm (**d**) Rapid response during two successive cycles of exposure to ammonia for 2 min and air flow for 3 min (100 ppm). The black arrows indicate the exposure gas type.

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
