# Peer review of "Direct Patterning and Spontaneous Self-Assembly of Graphene Oxide via Electrohydrodynamic Jet Printing for Energy Storage and Sensing"

_micromachines, 2019, doi:10.3390/mi11010013_

Round 1

Reviewer 1 Report

This manuscript reports using EHd printing for patterning laminar graphene oxide structure with self-assembly. The result is interesting. However, there are several places need to be clarified and corrected before publication. 

In the introduction, authors mentioned three processes or fabricating graphene-based films, but in the the later section during the comparing, authors used their results to compare with the result only from one process (direct dispensing method). Are there any particular reasons for not comparing their results with other processes?

For figure 1, if authors can show a picture of the poor alignment GO product fabricated by other methods will help readers to compare those two results. 

Equation 1, what is Θ? For equations 2 and 3, one of them has to be corrected to TV.

The ID of the nozzle is 100 micros if I am right, but what we got from figure 2 is that the resolution of the printed features is above 100 micros at most of the time. The advantage of EHD printing is creating high resolution feature that smaller than the nozzle size, but here the feature size is about or above the nozzle size. What is the possible reason to explain this result?  

Section 2.2, it mentioned that the nozzle was positioned 4mm above the silicon wafer. But in section 3.2 the working distance was changed to 10mm. Which one is correct? 

Section 3.3, it seems that the authors did not observe the coffee-ring effect in the EHD printed feature. It is very easy to see the coffee-ring effect in the printed liquid feather, but why the printed feature did not have the coffee-ring?

Section 3.3, 7.03 x 103 and 2.01 x 104 should be changed to 7.03 x 103 and 2.01 x 104.

For the printed ammonia sensor, did authors set a comparison group? If I am right, authors only test the performance of the printed sensor, but a commercial or similar sensor should be added to the experiment as a reference or comparison group. 

Reviewer 2 Report

The authors present a system for  LBL self-assembly of graphene oxide (GO) flakes using the EHD jet printing. The alignment of the GO flakes is hydrodynamically induced during the stretching of the EHD jet fibers managing the typical process parameters and the evaporation of the supernatant after deposition.

Micromachines could consider it as a potential publication.

Overall, this manuscript was well organized, and the characterizations were well performed. Major comments were listed as:

 One point that could be improved is the section of Introduction. The Introduction is too short, in particular, unconventional jet printing techniques capable of making direct patterns should be mentioned. I suggest the authors to add several sentences to explain the importance of this field. Some typical references listed here should be cited. Soft matter, 2016, 12 (25), 5542-5550 ; ACS applied materials & interfaces, 2019, 11 (3), 3382-3387; Nano Lett. 2011, 11 (4); Nano letters, 2007, 7(10), 3007-3012; Nature communications, 2012, 3, 890.

There is no information about the thickness of the jetted fibers, how does the thickness of the fibers change by changing the printing parameters? Have you had any nozzle clogging problems? how long does the printing take before the nozzle gets clogged?

The roughness can influence the wettability of the substrate and therefore the hydrophilicity/hydrophobicity of the substrate and therefore also the electrical characteristics and its stability. I suggest taking contact angle measurements

Reviewer 3 Report

1. Please put the mixing ratio of PVA and DIW on page 3, line 102.
2. You mentioned that you measured electrical resistance with a probe station on page 3, line 129.
Probe stations are instruments for helping your measurement.
When measuring electrical characteristics, it is measured by Source meter or Digital multi meter.
Please check the measuring instrument.
3. Why did you use r-GO to sense ammonia?
Please describe the mechanism of sensor based on r-GO for sensing ammonia.
4. Please mention which sensor you used in Figures 6c and d on page 10. (aspect ratio of 5 or 25??)
5. For sensors, the saturation curve contains a lot of information such as response time and On/Off ratio.
But Figures 6a and 6b only show the changes in resistance of an r-GO based 3D sensor.
How long does it take to get a saturation curve in an ammonia gas flow?
Please add the data showing the saturation curve, and describe the reponse speed and the On/Off ratio.
6. There is a mention of energy storage in the title of the this article, but the body of the article has no data as well as a lack of explanation.
It is recommended to move the data related to the supercapacitor in the supplementary to the main text or delete 'energy storage' from the title.
7. Please check the subscript and superscript. There are so many errors.
(For example, NH3 on page 3, line 138, 7.62 x 10-2 on page 7, 259 line, etc.)
8. You mentioned that the color changes of the pattern depend on the thickness of GO flakes on page 6, line 222.
If you have data or image for this, please include it in the supplementary.
9. In Figure 4, replace R-GO with r-GO.
10. Symbols for physical quantities (including constants) are italic letters, as are symbols for functions ingeneral.
However, in general, symbols used as subscripts and superscripts are roman.
Pleas check all of equations and symbols in this article.
11. Please check the journal's instruction for authors for references.

Round 2

Reviewer 1 Report

Authors have addressed all of my questions, and I recommend this manuscript for publication. 

Reviewer 2 Report

Thank you for making the requested revision. At the present form, I judge the manuscript suitable for the pubblication on Micromachines.  

Reviewer 3 Report

Most of them have been modified well, but I think it will be a good paper if you take care of them.

1.Please check again the subscript and superscript. (line 1925~ 196)

2.Please check again the journal's instruction for authors for references.

According to the instruction for authors of this journal, the guidelines for reference are as follows:

In the text, reference numbers should be placed in square brackets [ ], and placed before the punctuation; for example [1], [1–3] or [1,3]. For embedded citations in the text with pagination, use both parentheses and brackets to indicate the reference number and page numbers; for example [5] (p. 10). or [6] (pp. 101–105).
The reference list should include the full title, as recommended by the ACS style guide. Style files for Endnote and Zotero are available.
References should be described as follows, depending on the type of work:
Journal Articles:
1. Author 1, A.B.; Author 2, C.D. Title of the article. Abbreviated Journal Name Year, Volume, page range.
Books and Book Chapters:
2. Author 1, A.; Author 2, B. Book Title, 3rd ed.; Publisher: Publisher Location, Country, Year; pp. 154–196.
3. Author 1, A.; Author 2, B. Title of the chapter. In Book Title, 2nd ed.; Editor 1, A., Editor 2, B., Eds.; Publisher: Publisher Location, Country, Year; Volume 3, pp. 154–196.
Unpublished work, submitted work, personal communication:
4. Author 1, A.B.; Author 2, C. Title of Unpublished Work. status (unpublished; manuscript in preparation).
5. Author 1, A.B.; Author 2, C. Title of Unpublished Work. Abbreviated Journal Name stage of publication (under review; accepted; in press).
6. Author 1, A.B. (University, City, State, Country); Author 2, C. (Institute, City, State, Country). Personal communication, Year.
Conference Proceedings:
7. Author 1, A.B.; Author 2, C.D.; Author 3, E.F. Title of Presentation. In Title of the Collected Work (if available), Proceedings of the Name of the Conference, Location of Conference, Country, Date of Conference; Editor 1, Editor 2, Eds. (if available); Publisher: City, Country, Year (if available); Abstract Number (optional), Pagination (optional).
Thesis:
8. Author 1, A.B. Title of Thesis. Level of Thesis, Degree-Granting University, Location of University, Date of Completion.
Websites:
9. Title of Site. Available online: URL (accessed on Day Month Year).
Unlike published works, websites may change over time or disappear, so we encourage you create an archive of the cited website using a service such as WebCite. Archived websites should be cited using the link provided as follows:
10. Title of Site. URL (archived on Day Month Year).